# Dual GSK-3β/HDAC Inhibitors Enhance the Efficacy of Macrophages to Control *Mycobacterium tuberculosis* Infection

**DOI:** 10.3390/biom15040550

**Published:** 2025-04-09

**Authors:** Sadaf Kalsum, Ruilan Xu, Mira Akber, Shengjie Huang, Maria Lerm, Yuqing Chen, Magda Lourda, Yang Zhou, Susanna Brighenti

**Affiliations:** 1Center for Infectious Medicine (CIM), Department of Medicine Huddinge, Karolinska Institutet, ANA Futura, 141 52 Huddinge, Sweden; sadaf.kalsum@liu.se (S.K.); ruilan.xu@outlook.com (R.X.); mira.akber@ki.se (M.A.); 2Division of Medical Microbiology and Molecular Medicine, Department of Clinical and Experimental Medicine, Linköping University, 581 83 Linköping, Sweden; maria.lerm@liu.se; 3State Key Laboratory of Bioactive Molecules and Druggability Assessment, Guangdong Basic Research Center of Excellence for Natural Bioactive Molecules and Discovery of Innovative Drugs, International Cooperative Laboratory of Traditional Chinese Medicine Modernization and Innovative Drug Discovery of Chinese Ministry of Education School of Pharmacy, Jinan University, 855 Xingye Avenue, Guangzhou 510632, China; sjhuang@stu.jnu.edu.cn (S.H.); yuqing_chen2025@yeah.net (Y.C.); 4Division of Clinical Microbiology, Department of Laboratory Medicine, Karolinska Institutet, ANA Futura, 141 52 Huddinge, Sweden; magdalini.lourda@ki.se

**Keywords:** tuberculosis, *Mycobacterium tuberculosis*, macrophage, host-directed therapy, compound, glycogen synthase kinase 3 beta (GSK-3β), histone deacetylase (HDAC) inhibitors, inflammation

## Abstract

Multitarget drug discovery, including host-directed therapy, is particularly promising for tuberculosis (TB) due to the resilience of *Mycobacterium tuberculosis* (Mtb) as well as the complexity of the host’s immune response. In this proof-of-concept study, we used high-content imaging to test a novel panel of dual glycogen synthase kinase 3 beta (GSK-3β) and histone deacetylase (HDAC) 1 and 6 inhibitor candidates for their efficacy in reducing the growth of green fluorescent protein (GFP)-expressing mycobacteria in human primary macrophages. We demonstrate that all ten test compounds, also including the GSK-3β inhibitor SB415286, exhibit an antimycobacterial effect of 20–60% at low micromolar doses and are non-toxic to host cells. Mtb growth showed a positive correlation with the respective 50% inhibitory concentration (IC50) values of GSK-3β, HDAC1, and HDAC6 in each compound, indicating that compounds with a potent IC50 value for HDAC1, in particular, corresponded to higher antimycobacterial activity. Furthermore, the results from multiparametric flow cytometry and a customized multiplex RNA array demonstrated that SB415286 and selected compounds, C02 and C06, could modulate immune polarization and inflammation in Mtb-infected macrophages involving an enhanced expression of CCL2, IL-10 and S100A9, but a decrease in inflammatory mediators including COX-2, TNF-α, and NFκB. These data suggest that GSK-3β inhibition alone can decrease the intracellular growth of mycobacteria and regulate macrophage inflammation, while dual GSK-3β/HDAC inhibitors enhance this efficacy. Accordingly, the tailored design of dual GSK-3β/HDAC inhibitors could represent an innovative approach to host-directed therapy in TB.

## 1. Introduction

Despite worldwide efforts to fight tuberculosis (TB), it continues to be one of the leading causes of death from infectious diseases globally. In 2024, the WHO reported that around 10.6 million people worldwide suffer from TB, in addition to 1.2 million TB-related deaths [1]. Intracellular *Mycobacterium tuberculosis* (Mtb) infection requires daily treatment with multiple antibiotics for 6–9 months or longer [2]. Consequently, inappropriate use of chemotherapy creates problems with antibiotic resistance of Mtb to most known antibiotics, leading to reduced treatment success rates and accelerations in the dissemination of multidrug-resistant TB (MDR-TB) with around 400,000 new cases every year [1].

The multitargeting approach for developing novel anti-TB drugs has gained significant interest in recent years. The primary objective is to find new drugs that can simultaneously affect various pathways or mechanisms involved in the disease, enhancing therapeutic efficacy and decreasing the risks of drug resistance [3]. This includes host-directed therapies that are designed to achieve two main goals: reducing the pathological inflammation linked to TB disease progression and/or restoring antimicrobial immunity [4,5,6]. Here, epigenetic modulation with small chemical molecules, including histone deacetylase (HDAC) inhibitors, can be useful to alter immune cell differentiation and function [7,8]. Human cells have 18 HDAC proteins, grouped into four classes based on localization and activity [9]. The basic principle is that HDAC enzymes remove acetyl groups from histones, leading to tighter DNA wrapping and transcriptional repression [10]. HDAC inhibitors, classified into five structural groups, block HDACs to maintain histone acetylation, which typically results in an altered gene expression profile of the cell [11]. As such, HDAC inhibitors of different specificities may impact both innate and adaptive immune response genes, including inflammatory signaling pathways [12,13,14]. Therefore, numerous epigenetic modulators, including HDAC inhibitors, are currently being tested in preclinical studies and clinical trials for different diseases [15,16]. We have demonstrated in Mtb-infection models [17,18] and in randomized controlled trials [19,20] that the HDAC inhibitor, phenylbutyrate (PBA), can counteract the Mtb-induced manipulation of antimicrobial responses that inhibit Mtb growth in human macrophages and consequently improve the clinical outcomes of pulmonary TB in combination with antibiotics. PBA has a broad spectrum and inhibits class I (HDAC1, 2, 3, 8), class IIa (HDAC4, 5, 7, 9), and class IIb (HDAC6, 10) HDAC proteins [21]. Besides its protective effects in both bacterial [22,23,24,25] and viral infections [26,27] as well as cancer [28], PBA exerts neuroprotective effects and has been found to ameliorate cognitive deficit and reduce tau pathology in an Alzheimer’s disease (AD) model [29,30]. Interestingly, glycogen synthase kinase 3 beta (GSK-3β) is a protein kinase involved in various cellular processes and functions as a regulatory switch in host immune responses [31,32]. The inhibition of GSK-3β control Mtb infection in human macrophages was recently shown [33]. Likewise, GSK-3β plays a central role in AD due to its involvement in several key pathological processes, such as neuronal cell death and elevated inflammation [34]. Overall, these studies suggest that HDAC inhibitors, as well as GSK-3β inhibitors, could be utilized to modulate the antimicrobial and inflammatory pathways involved in human diseases.

In this proof-of-concept study, the objective was to evaluate a novel panel of dual GSK-3β and HDAC1/HDAC6 inhibitors for their potential to alter the growth of intracellular Mtb by modulating macrophage function. Originally designed to treat neurodegenerative disorders such as AD [35], we studied these compounds to determine their ability to reprogram immunity in Mtb-infected macrophages, aiming to inhibit bacterial growth. Our findings suggest that most dual GSK-3β-HDAC1/6 inhibitor candidates are non-toxic to host cells and can effectively reduce the intracellular growth of mycobacteria in the micromolar range, likely by modulating macrophage inflammation. These findings present new opportunities to further exploit dual GSK-3β/HDAC inhibitors to enhance the ability of human host cells to restrict Mtb growth.

## 2. Materials and Methods

### 2.1. Differentiation of Human Monocyte-Derived Macrophages (hMDMs)

This study used monocytes obtained from the buffy-coat blood of healthy adult volunteers enrolled at the Blood Center located at the Karolinska Hospital in Stockholm, Sweden, with informed consent. Ethical approval was obtained from the Ethical Review Board in Stockholm (EPN), Sweden (2010/603-31/4).

Lymphoprep™ (Alere technologies, Oslo, Norway) was used to separate peripheral blood mononuclear cells (PBMCs) that were cultured in T75 flasks (Corning, NY, USA), and monocytes were isolated using plastic adherence for 2–3 h at 37 °C in serum-free RPMI 1640 media (VWR, Radnor, PA, USA). Around 70 million PBMCs were added to each T75 flask, corresponding to approximately 7 million monocytes (10% of total PBMCs). Next, adherent monocytes were differentiated into macrophages using 50 ng/mL of the macrophage colony-stimulating factor (M-CSF from PeproTech, Cranbury, NJ, USA) added to the cell culture medium (RPMI supplemented with 10% fetal calf serum (FCS from VWR, Radnor, PA, USA), 2 mM L-glutamine, 1 mM sodium pyruvate, and 5 mM HEPES from Cytiva, Marlborough, MA, USA), i.e., 10 mL/flask for 7 days. On day 3, half of the cell culture medium was replaced with fresh media containing M-CSF for another four days to obtain fully differentiated human monocyte-derived macrophages (hMDMs). The removal of adherent hMDMs was performed using a 2 mM EDTA (Sigma Aldrich, Saint Louis, MO, USA) buffer with 2.5% FCS for 30 min at 37 °C, which typically resulted in >98% cell viability. hMDMs were washed and re-seeded at appropriate cell numbers for experiments in microtiter plates as described below.

### 2.2. Preparation of Mycobacterial Cultures

Avirulent H37Ra and virulent H37Rv mycobacteria that were transfected with a plasmid carrying a gene-encoding green fluorescent protein (GFP; pFPV2) were obtained from ATCC (Rockville, MD, USA). Mycobacteria were sub-cultured in complete TB media (Middlebrook 7H9, 10% OADC supplement, 0.05% Tween-80, and 20 µg/mL kanamycin) obtained from the Karolinska University Hospital, Stockholm, Sweden. Typically, mycobacterial cultures were maintained at 37 °C for 2 weeks before washing in PBS-Tween-80 (Sigma Aldrich, Saint Louis, MO, USA) and incubating the cultures for an additional week. After washing in 0.05%PBS-Tween-80, the bacterial suspension was sonicated for 5 min using the pulse-chase method to disrupt the bacterial clumps before optical density was assessed at 600 nm and adjusted to 0.2–0.3. Bacterial aliquots were immediately used for the infection of hMDMs or kept at −80 °C in 7H9 medium diluted with 35% glycerol (both from Sigma Aldrich, Saint Louis, MO, USA) for future experiments.

### 2.3. Mycobacterial Infection of hMDMs

For the mycobacterial infection of hMDMs, we used a previously established protocol with some modifications [18,36,37]. Accordingly, human macrophages were infected with mycobacteria in 96-well (10,000 hMDMs/well), 384-well (4000 hMDMs/well) or 6-well (1 million hMDMs/well) plates as follows: (1) H37Ra was used at multiplicity of infection (MOI) 5 after sub-culture in 7H9-supplemented media for 7–10 days at 37 °C. After 4 h of infection, extracellular mycobacteria were washed off, and complete cell culture medium was added with or without test compounds. (2) H37Rv was used at an MOI of 1 straightaway after the thawing of bacterial aliquots stored at −80 °C. Mtb-infected cell cultures were not washed, but mycobacteria and the test compounds were added at the same time. All test or control compounds were used in triplicates for 96- or 384-well plates. Plates were incubated at 37 °C for 4 h (flow cytometry), 24 h (RNA expression analyses), or 3–5 days (Mtb growth and cell viability).

### 2.4. Flow Cytometry

Flow cytometry was used to evaluate the proportion of the monocyte population in PBMCs from healthy donors before differentiating them to hMDMs and after M-CSF-mediated differentiation to hMDMs. The phenotype of hMDMs before and 4 h after Mtb infection (MOI 1) was also evaluated with flow cytometry. Untreated Mtb-infected hMDMs cultured in a medium with a diluent only were used as positive controls, while uninfected hMDMs were used as negative controls. For either PBMC or hMDM staining, the cells were detached and washed with the FACS buffer (PBS supplemented with 0.5 mM EDTA and 5% FCS), followed by a 15 min incubation at room temperature in the dark with a 50 µL antibody cocktail (all antibodies were obtained from Biolegend) containing (i) mouse anti-human CD14 (BV570) and LIVE/DEAD Near IR Viability Dye (Invitrogen, Waltham, MA, USA) for PBMCs and (ii) Fc-block (Miltenyi, Bergisch Gladbach, Germany), LIVE/DEAD Viability Dye Zombie-UV (Invitrogen) CD45 (AF700), TLR2 (AF647), HLA-DR (PE-Cy5), CD64 (PE/DAZZLE), CD200R (PE), CD86 (BV785), CD80 (BV650), CCR7 (BV711), CD163 (BV605), and CD206 (APC-Cy7) for hMDMs. H37Ra-GFP bacteria were detected in the FITC channel. After staining, cells were fixed with a fixation/permeabilization working solution (BD Cytofix/Cytoperm™, Franklin Lakes, NJ, USA), followed by 10 min intracellular staining with mouse anti-human CD68 (PE-Cy7). After washing, stained cell samples were resuspended in the FACS buffer and acquired either on an LSR Fortessa (BD Biosciences, Franklin Lakes, NJ, USA; PBMCs) or on a BD FACSymphony A3 (BD Biosciences, Franklin Lakes, NJ, USA; hMDMs), equipped with five lasers. Gating strategies included the exclusion of doublets and dead cells. The fluorescence intensity of markers was visualized in histograms as the median. All stained samples were acquired at a low-to-medium speed.

### 2.5. Test and Control Compounds

In the panel of test compounds, nine candidates were designed to inhibit the activities of GSK-3β and HDAC1/HDAC6 enzymes, while one compound inhibited GSK-3β only. These compounds were kindly provided by Yang Zhou at Jinan University, Guangzhou, China. SB415286 is a commercially available GSK-3β inhibitor with a 50% inhibitory concentration (IC50) of 78 nM that was synthesized according to procedures previously described [38,39]. The test compounds had a molecular weight between 350 and 600 g/mol. The IC50 values of compounds C01-C09 determined against the various targets, GSK3β, HDAC1, and HDAC6, were assessed by WuXi AppTec (Shanghai, China). Half of the dual GSK-3β/HDAC inhibitors had an IC50 value lower than SB415286 (C01, C02, C04, C05, C07), while four compounds had a higher IC50 than SB415286 (C03, C06, C08, C09).

In this study, dual GSK-3β/HDAC inhibitors (0.1–10 µM) were incubated with hMDM infected with GFP-expressing mycobacteria. Treatment with the medium only was used as a negative control, while treatments with the primary anti-TB drugs, rifampicin (RIF) and isoniazid (INH) (1 µg/mL) (Sigma Aldrich, Saint Louis, MO, USA), or the pan-HDAC inhibitor sodium phenylbutyrate (2 mM) (Cayman Chemical, Ann Arbor, MI, USA) were used as positive controls. All test and control conditions were diluted in the DMSO solvent (Sigma-Aldrich) to a final concentration of 0.1% in the cell culture media.

### 2.6. High-Content Imaging with IncuCyte

The growth of GFP-labeled mycobacteria inside hMDMs and the viability of infected hMDMs were monitored with the IncuCyte^®^ S3 Live-Cell Analysis System (Sartorius, Göttingen, Germany) for up to five days. Accordingly, intracellular mycobacteria were visualized in green light, while host cell viability was quantified using 250 nM of the Cytotox Red Reagent (Sartorius, Göttingen, Germany), which was added at the start of the cell cultures. Treatment with 4% formaldehyde or 10 mM suramin (Sigma Aldrich, Saint Louis, MO, USA) was used for positive controls to determine cytotoxicity. Image acquisition was based on 9 images/well in 96-well plates and 2 images/well in 384-well plates at 20× magnification taken every 12 h during the experiment. Alternatively, a single image of the whole well was captured using a 4× objective at a single time point on day 4 post-infection. The time point for readout depended on the rate of intracellular mycobacterial growth in the MOI control and the integrity of Mtb-infected macrophages in culture. If the bacteria were growing rapidly in cells from a given donor, the experiment was concluded on day 3. Conversely, bacteria that were growing slowly in the cells could be maintained until day 5 before the final readout. The IncuCyte S3 software was used to analyze the fluorescence as the total integrated intensity (GCU × µm^2^/image). Results are displayed as percentages (%) of intracellular Mtb growth in macrophages relative to the untreated MOI control (set to 100%) or as % cell viability relative to the uninfected and untreated control (set to 100%).

### 2.7. Colony Forming Unit (CFU) Counts

Standard CFU counts were used as a secondary assay to quantify Mtb growth in macrophages after exposure to test or control compounds. Infected cells were treated with a cell lysis buffer (0.036% SDS in Milli-Q water; SDS from Sigma Aldrich, Saint Louis, MO, USA) for 5 min on day 3. The cell lysates were diluted in PBS (1/10 to 1/10,000) and plated in duplicates on Middlebrook 7H10 agar plates containing 10% OADC (Karolinska University Hospital, Stockholm, Sweden). CFU counts were determined after 3 weeks of incubation of the plates at 37 °C. Results were presented as percentages (%) of intracellular Mtb growth in macrophages relative to the untreated MOI control (set to 100%).

### 2.8. TaqMan Array Cards for Quantitative PCR

The transcriptional profiling of macrophage inflammation was assessed at 24 h using multiplex RNA analyses. For this purpose, we used custom-made TaqMan Gene Expression Assays in 384-well TaqMan Array plates (Thermo Fisher Scientific, Waltham, MA, USA) including primers and probes for the detection of IL1β, IL-6, TNF-α, IL-10, IL-8, CCL2, CCL4, CCL5, TLR2, Atg 7, Atg14, Beclin-1, CAMP (LL-37), S100A9, NOS2 (iNOS), CYBB (NOX2), PTGS2 (COX2), NFκB, SOCS-3, NLPR3, Nrf2, STING, and TREM2. The housekeeping 18S rRNA was used as a reference gene. Uninfected or H37Rv-infected macrophages (MOI5) were used as baseline controls, while mRNA expression was quantified in cell cultures treated with control (cell culture medium or 2mM PBA) or test compounds (1 µM each of SB415286, C02 or C06). RNA was isolated from macrophage cultures using the Ribopure RNA extraction kit (Ambion, Thermo Fisher Scientific, Waltham, MA, USA), while TaqMan Fast Advanced Mix was used for qPCR amplification, and SuperScript IV VILO Master Mix was used for cDNA synthesis (Thermo Fisher Scientific). cDNA samples were assessed in triplicates using quantitative real-time PCR conducted at the BEA core facility (Bioinformatics and Expression Analysis) at Karolinska Institutet, Sweden. TaqMan arrays were assessed using the Applied Biosystems™ 7900HT Fast Real-Time PCR System with a 384-well block module (Waltham, MA, USA). Acquired data were analyzed using the relative standard method where cycle threshold (Ct) values for the target genes were normalized to the housekeeping gene 18S. Results were presented as the fold change in mRNA in the different conditions compared to uninfected and untreated control cells (the fold change was set to 1).

### 2.9. Statistics

Data were plotted and analyzed in GraphPad Prism 9 (version 9.5.1). Bar graphs presenting data from 4 to 6 donors in total were presented as the median and range (or interquartile range). Mycobacterial growth in macrophages was expressed as the percentage (%) relative to the untreated MOI1 control. The efficacy of the compounds was quantified as the percentage (%) of bacterial growth inhibition relative to the untreated MOI1 infection control. *p*-Values were determined using a two-way ANOVA and Dunnett’s or Sidak’s multiple comparisons test or Friedman’s test for >two groups. Spearman’s correlation test was used to evaluate the association between intracellular Mtb growth and the IC50 values of the respective enzymes in each compound. The correlation analysis was conducted and plotted using R (version 4.4.1). The value of r = 1 indicates a perfect positive correlation, whereas r = −1 indicates a perfect negative correlation. Statistical analyses resulting in a *p*-value < 0.05 * were considered significant.

## 3. Results

### 3.1. Phenotypic Screening of Dual GSK-3β and HDAC1/HDAC6 Inhibitor Candidates Using a Human Macrophage Infection Model

In this study, we tested the antimycobacterial efficacy of a panel consisting of nine dual GSK-3β and HDAC1/HDAC6 inhibitor (C01–C09) candidates and one GSK-3β inhibitor (SB415286) with different inhibitory profiles, i.e., IC50 values for the individual enzymes (unpublished observations). Compounds C01–C09 possessed variable IC50 values against GSK-3β (ranging from 12.1 to 520 nM), HDAC1 (ranging from 9.9 to 6076.3 nM), and HDAC6 (ranging from 1.9 to 8553.3 nM). All compounds inhibited GSK-3β and both HDAC1 and HDAC6, except for compound C09, which inhibited GSK-3β and HDAC1 only. We assessed the effectiveness of the compounds in reducing the growth of GFP-labeled mycobacteria inside human cells using an in vitro macrophage infection model [36,37] and high-content live-cell imaging with IncuCyte (Figure 1a).

First, the flow cytometry of uninfected cells was used to determine the percentage of monocytes among bulk PBMCs, which was found to be around 10% (Figure 1b, upper panel). The purity of macrophages that differentiated in M-CSF for 7 days was around 60%, including cells that were double-positive for the myeloid cell markers CD68 and CD14, with cell viability being high (>99%) (Figure 1b, lower panel). Some of the CD68- cells were CD14+ monocytes, while cells that were CD68-CD14- were likely contaminating lymphocytes (Figure 1b). Furthermore, the flow cytometry of H37Ra-infected macrophages at an MOI of one demonstrated that most infected cells were CD68 + CD14+, while the GFP signal was considerably weaker in CD68- cells (Figure 1c). Additionally, hMDMs infected with H37Ra at an MOI of one for 4 h had approximately 4.8% infected CD68 + CD14+ macrophages, compared to 68% infected cells at an MOI of five (Figure 1d). Accordingly, an MOI of one results in a low level of Mtb infection, whereas an MOI of five leads to a high level of Mtb infection.

### 3.2. Evaluating the Efficacy of Dual GSK-3β and HDAC1/HDAC6 Inhibitor Candidates in Reducing the Growth of Avirulent H37Ra in Human Macrophages

Different concentrations of test compounds were used to assess their anti-mycobacterial activity on hMDMs infected with H37Ra-GFP (Figure 2a). The results demonstrate that SB415286, as well as compound C01–C09, possess an inherent ability to restrict intracellular Mtb growth to a variable extent (Figure 2a). It should be noted that SB415286 was as potent as PBA in inhibiting the growth of Mtb in macrophages at the 10 µM dose (Figure 2a). Compared to the MOI1 infection control, we observed a significant reduction in bacterial growth in the presence of 10 µM of compounds C02 (*p* < 0.0004) and C06 (*p* < 0.004) (Figure 2a). Accordingly, the 10 µM dose was significantly more effective than the 1 µM dose of C02 (*p* < 0.03) and C06 (*p* < 0.006), demonstrating a clear dose–response effect (Figure 2a). Lower doses, including 0.01 and 0.1 µM, were also tested but did not have a major effect on intracellular Mtb growth compared to the control (unpublished observations). The donor variation and individual response to the compounds appeared relatively large, and some variability could also be detected in the presence of RIF and INH (Figure 2a). Host cell viability determined in the macrophage cultures treated with control or test compounds revealed no overt cytotoxicity induced in the presence of 1 or 10 µM concentrations (Figure 2b).

Next, H37Ra growth inhibition data in Figure 2a were used to quantify compound efficacy at the 10 µM dose (Figure 2c). From this analysis, we observed a significant efficacy of compound C02 (*p* < 0.004) and C06 (*p* < 0.028) to restrict intracellular Mtb growth in human host cells compared to the untreated control (Figure 2c). As expected, antibiotics, RIF + INH, were very effective in restricting in vitro Mtb growth (*p* < 0.0001) (Figure 2c). Interestingly, the GSK-3β inhibitor SB415286 exhibited efficacy comparable to that of the PBA control (Figure 2c). Compounds C02 and C06 had relatively low IC50 values for inhibiting HDAC1 and HDAC6 compared to the other test compounds, while their IC50 values toward GSK-3β were in a similar range as SB415286. Intracellular Mtb growth was further associated with the respective IC50 value of GSK-3β, HDAC1, or HDAC6 in each compound (Figure 2d). Spearman’s correlation analyses suggested a positive correlation between Mtb growth and the IC50 of all three enzymes, including a significant positive correlation (r = 0.77, *p* < 0.021) with HDAC1 (Figure 2d). These data indicate that compounds with a potent IC50 value for HDAC1 correspond to a higher antimycobacterial activity, which aligns with previous findings that Mtb infection considerably upregulates HDAC1 in human macrophages [40].

### 3.3. Evaluating the Efficacy of Dual GSK-3β and HDAC1/HDAC6 Inhibitor Candidates in Reducing the Growth of Virulent H37Rv in Human Macrophages

After evaluating the efficacy of the compounds using avirulent H37Ra mycobacteria, we proceeded to assess their efficacy and reduce the intracellular growth of virulent H37Rv. For these experiments, a slightly different experimental setup was used, as illustrated in Figure 1. This approach has been optimized for the medium throughput screening of compounds in 384-well plates. The efficacy of the individual test compounds in reducing the growth of H37Rv in macrophages appeared to be more potent in the lower dose range of 1 µM compared to that of H37Ra (Figure 2a and Figure 3a). All compounds demonstrated a significant reduction in H37Rv growth at the 1 and 10 µM doses compared to the MOI1 infection control (with *p*-values ranging from *p* < 0.02 to *p* < 0.0001), except for C01 at 1 µM and C03 at 10 µM (Figure 3a). Lower doses than 1 µM or the higher 10 µM dose (Figure 3a) did not improve the overall efficacy of the test compounds. Instead, intracellular bacterial growth increased (*p* < 0.0001) upon treatment of Mtb-infected cells with 10 µM of compounds C03 and C07 (Figure 3a). While the cell viability of H37Rv-infected cells treated with the test compounds was similarly low compared to H37Ra-infected cells (Figure 2b), the cell morphology appeared rounded and compromised at 10 µM in contrast to 1 µM of C03 and C07 (Figure 3b). Overall, the majority of dual inhibitors demonstrated potent inhibition of H37Rv growth in human host cells in comparison to MOI1 infection control, and the efficacy of compound C06 was significant at the 1 µM dose (*p* < 0.02) (Figure 3c).

We continued with standard CFU counts, including compounds C02 and C06, which were overall most effective in reducing H37Ra (Figure 2c) as well as H37Rv (Figure 3c) growth in macrophages. Here, the GSK-3β inhibitor SB415286 was included as a comparator. CFU results suggested that all three compounds, including SB415286, and especially C02 and C06, could reduce intracellular H37Rv growth in a dose-dependent manner (Figure 3d). Furthermore, flow cytometry demonstrated that the infectivity and uptake of Mtb-GFP bacteria inside hMDMs was not significantly altered by treatment with 10 µM SB415286, C02, or C06 compared to the MOI1 control or treatment with RIF + INH (Figure 3e).

Representative microscopy images of H37Rv-infected macrophages illustrate that bacterial growth was low in the presence of most compounds, while macrophage viability and cell integrity remained high at the 1 µM dose (Figure 3f).

### 3.4. Modulation of Mtb-Infected Macrophage Responses upon Treatment with the GSK-3β Inhibitor SB415286 and Compounds C02 and C06

To assess macrophage inflammation in response to Mtb infection and how this was modulated by dual GSK-3β/HDAC inhibitor candidates, we exploited multiparametric flow cytometry and a customized multiplex mRNA array using the TaqMan Gene Expression Assay. The differentiation and activation of uninfected and Mtb-infected macrophages were assessed using 12-color flow cytometry in the presence or absence of treatment with the selected compounds SB415286, C02, and C6 at 4 h (Figure 4). The histograms in Figure 4a demonstrate the changes in classical macrophage markers in response to Mtb infection as well as the test compounds. While CD68 expression did not change with infection or treatment, Mtb infection resulted in the elevated expression of HLA-DR and CD200R, while activation markers such as CCR7, CD80/CD86, and CD64 as well as scavenger receptors CD163 and CD206 were reduced on Mtb-infected cells (Figure 4a). The treatment of Mtb-infected macrophages with SB415286, as well as C02 or C06, resulted in the up-regulation of TLR2 and partially restored the expression of co-stimulatory molecules CD80 and CD86 (Figure 4a and unpublished observations). While the test compounds did not have a major effect on CD64 at this time point, we observed a clearly elevated expression of CD163, CD200R, and CD206 (Figure 4a,b). The difference in CD206 expression, when comparing the MOI1 control and compound C06, was significant (*p* < 0.04). No difference was detected comparing the GSK-3β inhibitor, SB415286, with the dual inhibitor candidates, C02 and C06 (Figure 4b). These data imply that the selected test compounds could alter the macrophage polarization of Mtb-infected cells.

The TaqMan array card included 23 genes associated with inflammatory responses and regulatory mechanisms in macrophages (Appendix A). The results from differentially regulated markers in uninfected and Mtb-infected cells in response to PBA or the test compounds SB415286, C02, and C6 at 24 h are shown in Figure 5.

Mtb infection significantly influenced the expression of various markers in the macrophages compared to uninfected control cells (Figure 5). Infection with Mtb and treatment with the test compounds primarily affected pro-inflammatory cytokines/chemokines (Figure 5a–f) or regulators of inflammation and macrophage effector functions (Figure 5g–l). Pro-inflammatory gene expression was generally elevated in Mtb-infected cells, including IL-1β, IL-6, IL-8, TNF-α, CCL4, CCL5, and NFκB, but also the suppressor of cytokine signaling 3 (SOCS-3) and prostaglandin-endoperoxide synthase 2 (PTGS2) encoding the inducible cyclooxygenase (COX)-2 isoform (Figure 5). PBA and the test compounds affected gene expression in Mtb-infected cells to varying degrees (Figure 5). The most prominent modifications induced by the test compounds included the enhanced expression of CCL2 (*p* < 0.04), anti-inflammatory IL-10 (*p* < 0.04), and the antimicrobial protein S100A9 (Figure 5d,i,k), with a concomitant decrease in COX-2 (*p* < 0.02), TNF-α, and NFκB (Figure 5h,j,l). Overall, the results from the flow cytometry of surface markers, as well as this RNA array, suggest that it is possible to modify the inflammation of Mtb-infected macrophages with dual GSK-3β/HDAC inhibitor candidates.

## 4. Discussion

Multitarget drug discovery is a promising approach for treating complex diseases, such as cancer, neurological disorders, and chronic infections, which involve multiple pathways and mechanisms [41]. In this proof-of-concept study, we took advantage of an existing panel of dual GSK-3β/HDAC inhibitor candidates generated for the treatment of AD and demonstrated that these compounds could reduce the growth of Mtb inside human macrophages. All ten compounds, including the GSK-3β inhibitor SB415286, showed an antimycobacterial effect of 21–62% at low doses in the micromolar range. Strikingly, SB415286 exhibited similar efficacy and restricted intracellular Mtb growth compared to the HDAC inhibitor, PBA. Our observations particularly highlight the potent efficacy of two compounds, C02 and C06, in reducing the growth of both avirulent and virulent mycobacteria inside human macrophages. These compounds had relatively low IC50 values for inhibiting HDAC1 and HDAC6 compared to the other test compounds. None of the test compounds were toxic to Mtb-infected macrophages; however, C03 and C07 appeared to alter the morphology of Mtb-infected macrophages at the 10 µM dose. Furthermore, the test compounds demonstrated the ability to modulate inflammation in Mtb-infected macrophages by altering immune polarization and including an enhanced expression of CCL2, IL-10, and S100A9, but also decreasing inflammatory mediators such as COX-2, TNF-α, and NFκB. Overall, these data demonstrate that GSK-3β inhibition alone can reduce intracellular Mtb growth and modulate macrophage inflammation, while dual GSK-3β/HDAC inhibitors enhance this efficacy.

In this study, we took advantage of a well-established in vitro infection model using primary hMDMs that we developed for studies on immunomodulation and host-directed therapies in TB [36,37]. This model is adaptable to various microplate formats and employs different MOIs based on specific scientific questions and/or infection conditions. Here, we used slightly different experimental setups to assess the growth inhibition of avirulent and virulent mycobacteria in the presence of compounds. Overall, compound efficacy ranged from 21 to 50% at the higher 10 µM dose using the H37Ra assay at MOI5 compared to 29–62% at the lower 1 µM dose using the H37Rv assay at MOI1. As such, it appears more challenging to restrict intracellular Mtb growth in macrophages infected with a higher MOI of five compared to an MOI of one, which may represent a more physiologically relevant condition [42]. It is likely that a low MOI more closely mimics natural infection and in vivo situations, where macrophages encounter lower numbers of bacteria. In contrast, higher MOIs may lead to overwhelming infection, making it more difficult for macrophages to control Mtb. Nonetheless, both assay formats demonstrated high efficacy of the compounds, particularly C02 and C06, reinforcing the conclusion that dual GSK3β-HDAC1/6 inhibitor candidates effectively support the restriction of intracellular Mtb growth in human cells.

It is well described that chronic inflammation is associated with progressive TB disease, which involves multiple cytokines and mediators of inflammatory pathways [43,44,45,46]. GSK-3, composed of two closely related isoforms, GSK-3α and GSK-3β, is a powerful driver of inflammation in different disease contexts, making GSK-3 inhibitors a promising focus for anti-inflammatory research [47]. Accordingly, GSK-3 inhibitors have been shown to modulate inflammatory response by increasing IL-10 production and suppressing the NFkB-mediated activation of pro-inflammatory cytokines [48]. Likewise, GSK-3β has been shown to act as a molecular switch for the regulation of the cytokine milieu in Mtb-infected dendritic cells (DCs), and the inhibition of GSK-3β resulted in lower IL-1β but elevated IL-10 levels [49]. Our results suggested that GSK-3β inhibition with SB415286 could skew early macrophage activation towards an anti-inflammatory phenotype featured by reduced CD64 but elevated CD163, CD200R, and CD206 expressions. Furthermore, multiplex RNA array data imply that SB415286, as well as dual GSK-3β/HDAC inhibitor candidates, C02 and C06, can modify the Mtb-induced activation of pro-inflammatory cytokines and chemokines and increase IL-10 but decrease COX-2 expression in Mtb-infected cells. Previous studies have shown that COX-2 plays a significant role in TB pathogenesis and is involved in the production of prostaglandin E2 (PGE2), which is a lipid compound that modulates inflammation [50]. The inhibition of GSK-3 reduces PGE2 production by decreasing COX-2 expression in monocyte/macrophage lineage cells [51]. Consistently, the Mtb infection of human macrophages triggers a strong NFkB-mediated inflammatory response, including the upregulation of COX-2 along with substantial PGE2 production that can be prevented by the treatment of cells with anti-inflammatory drugs [52]. It has also been shown that COX-2 inhibition reduces PGE2 production, which limits INH- and RIF-tolerant Mtb populations residing in mesenchymal stem cells [53]. In contrast, studies using plasma and blood cells from TB patients suggest that PGE2 has a potent immunosuppressive role on innate and adaptive immune responses in Mtb infection, potentially protecting the host from excessive inflammation [54]. However, an indirect reduction in COX-2 via GSK-3β inhibition or dual GSK-3β/HDAC inhibition may balance the immune response to prevent pathological inflammation while avoiding excessive immunosuppression.

The specific involvement of GSK-3β in AD has been studied for several decades, and the therapeutic potential of GSK-3β inhibitors as potential disease-modifying agents is currently under investigation [55,56]. This also involves the design of GSK-3β-based dual inhibitors such as the combination of GSK-3β and HDAC1 and 6 [35] or HDAC2 and 6 [57] with immunomodulatory properties. Such dual GSK-3β/HDAC inhibitor candidates have also been generated using computational modeling [58]. Lately, the roles of GSK-3β and HDACs in regulating protective immunity to TB have garnered increased attention. A recent study screened hundreds of kinase inhibitor compounds and identified several specific GSK-3β inhibitors that effectively reduced intracellular Mtb growth in THP-1 cells at concentrations below 10 µM [33]. This finding was confirmed by siRNA downregulation and the CRISPR inactivation of GSK-3β in THP-1 macrophages, which resulted in an approximately 30–50% inhibition of intracellular Mtb growth [33]. Interestingly, the GSK-3β inhibitor compounds appeared to exert their activity on cells infected with mycobacteria expressing the protein tyrosine phosphatase (PtpA) [33], which is a virulence protein known to interfere with antimicrobial responses in macrophages, including the prevention of phagosomal acidification [59,60]. Accordingly, a microarray analysis of phosphosites in human signaling proteins from Mtb-infected THP-1 cells treated with a selective GSK-3β inhibitor, P-4423632, identified proteins linking GSK-3β to both phagocytosis and host cell apoptosis [33]. While the precise mechanisms of the GSK-3β-mediated growth control of Mtb in human macrophages remain to be determined, this report supports our results and warrants the further investigation of GSK-3β inhibition in the presence or absence of HDAC inhibition in other experimental models.

Many diseases are caused by a combination of factors, making treatment with drugs that target a single molecule difficult. Multitarget drugs may be more effective in treating complex diseases such as TB and AD and could also reduce the likelihood of drug resistance [41,61]. Importantly, such drugs can potentially reduce side effects by targeting multiple pathways at lower doses. Accordingly, we previously demonstrated that a combination of PBA + vitamin D and rifampicin or isoniazid was as effective in reducing MDR-TB growth in human macrophages as a 65- up to a 125-fold higher dose of antibiotics alone [18], demonstrating the potent additive effects of immunomodulatory treatment and antibiotics. Consistent with these findings, a recent study suggested that selected HDAC inhibitors could enhance the susceptibility of MDR-TB isolates in macrophages to rifampicin [62]. Furthermore, we recently demonstrated that a specific group of HDAC inhibitors targeting sirtuins could reduce intracellular Mtb growth in macrophages by 45–75% at micromolar doses, compared to an average reduction of 40% for PBA at a fixed 2 mM dose [63]. These HDAC inhibitors, including tenovin, suramin, salermide, and cambinol, primarily appeared to inhibit sirtuin 2, and antimicrobial synergy testing revealed additive effects between the sirtuin inhibitors and subinhibitory concentrations of rifampicin or isoniazid [63]. Around half of 20 commercially available HDAC inhibitors with different specificities, tested for their ability to reduce intracellular Mtb growth, were found to be cytotoxic to Mtb-infected cells at a 1 µM dose [63], which contrasts with the dual GSK-3β-HDAC1/6 inhibitors explored in this study. These compounds did not appear to be toxic to host cells using micromolar doses, but two compounds, C03 and C07, altered cellular morphology at the higher 10 µM dose without reducing membrane integrity, which may indicate differential macrophage polarization. As such, it has been proposed that pro-inflammatory M1 macrophages are round, and anti-inflammatory M2 macrophages are elongated [64], but this morphological change may also represent pro-apoptotic or early apoptotic cells that are impermeable to viability dyes. In our previous study, most cytotoxic HDAC inhibitors were broad-spectrum compounds [63], suggesting that more specific HDAC inhibition may be desirable for host-directed therapy in TB. From this perspective, it would be interesting to explore if the computational design of dual GSK-3β/sirtuin inhibitors could provide additional benefits by enhancing immune cell polarization and function, thereby improving disease outcomes in TB alongside conventional antibiotics.

## 5. Conclusions

Our results provide evidence that dual GSK-3β/HDAC inhibitor candidates with different IC50 values can control the intracellular growth of Mtb to a variable extent by modulating macrophage inflammation and function. These findings pave the way for the tailored design of dual GSK-3β/HDAC inhibitors that can be developed as a form of host-directed therapy for TB.

## Figures and Tables

**Figure 1 biomolecules-15-00550-f001:**
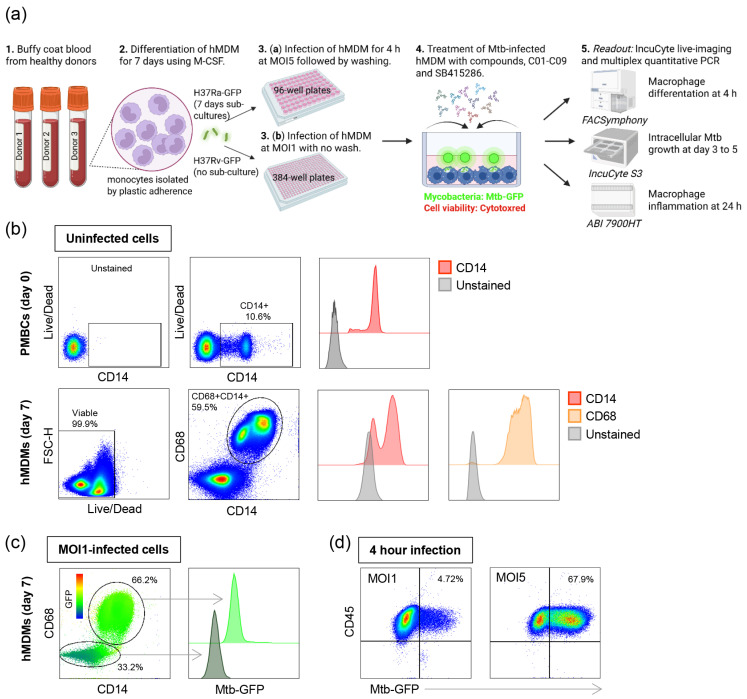
In vitro Mtb infection model using hMDMs. (**a**) Schematic illustration describing the experimental study outline. (**b**) Representative flow cytometry plots of uninfected hMDMs showing cell viability and the percentage of CD14+ monocytes among PBMCs (upper panel) as well as cell viability and the percentage of double-positive CD68+CD14+ macrophages after differentiation in M-CSF for 7 days (lower panel). Histograms showing the median expression of CD14 (red) and CD68 (brown) compared to unstained control cells (light gray). (**c**) Representative flow cytometry plot of H37Ra-infected hMDMs showing the intensity of GFP-labeled bacteria inside the CD68+CD14+ and the CD68-negative cell populations after 4 h of infection. Histogram showing the median expression of Mtb-GFP in CD68+CD14+ macrophages (light green) and CD68-negative cells (gray-green). The color key from blue to red indicates low-to-high expression levels of GFP. (**d**) Representative flow cytometry plots of H37Ra-infected CD68+CD14+CD45+ hMDMs at MOI1 and MOI5 showing the percentage of GFP-positive macrophages in the upper right squares.

**Figure 2 biomolecules-15-00550-f002:**
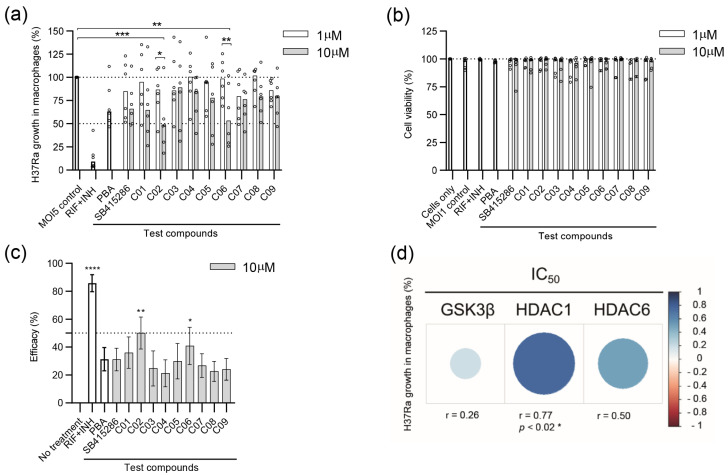
Efficacy of SB415286 and the dual GSK-3β and HDAC1/HDAC6 inhibitor candidates to reduce the growth of H37Ra-GFP in human macrophages. (**a**) Intracellular H37Ra growth was assessed with high-content imaging using 1 and 10 µM doses of the ten compounds SB415286 and C01-C09. The dotted lines show 50% and 100% (MOI5 infection control) intracellular growth of Mtb. The control groups included fixed doses of RIF + INH (1 µg/mL) or PBA (2 mM), which are highlighted as bold bars. Data in the scatter dot plot (n = 6 donors) show the median. The difference between 1 and 10 µM concentrations was determined using a two-way ANOVA and Sidak’s multiple comparisons test, while the difference between the MOI5 infection control and the test compounds was determined using a two-way ANOVA and Dunnett’s multiple comparisons test, *** *p* < 0.001, ** *p* < 0.01, * *p* < 0.05. (**b**) The host cell, i.e., macrophage viability, was assessed at 1 and 10 µM doses of the ten compounds. Data in the scatter dot plot (n = 6 donors) show the median. Untreated and uninfected cells were only set to 100% viability. The control groups are highlighted as bold bars. (**c**) Efficacy presented as the intracellular Mtb growth inhibition (%) of the controls and the test compounds, SB415286 and C01-C09, was determined at 10 µM. The control groups included fixed doses of RIF + INH or PBA, which are highlighted as bold bars. Data in the bar graph (n = 6 donors) show the median +/− interquartile range and the difference in efficacy compared to untreated H37Ra-infected cells was determined using Friedman’s test, **** *p* < 0.0001, ** *p* < 0.01, * *p* < 0.05. (**d**) Heat map depicting the correlation between intracellular Mtb growth and the respective IC50 value (1–10.000 nM) of the individual enzymes in each compound. Blue indicates a positive correlation, while red colors reflect the strength of a negative correlation. The size of the dots represents the strength of the correlation determined using Spearman’s correlation test, * *p* < 0.05.

**Figure 3 biomolecules-15-00550-f003:**
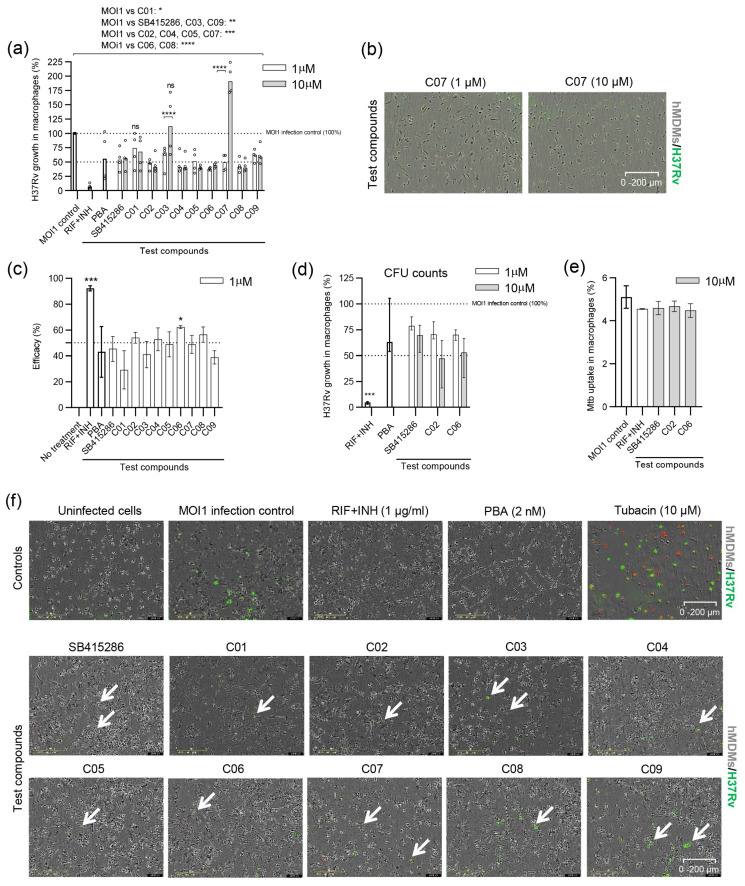
Efficacy of SB415286 and the dual GSK-3β and HDAC1/HDAC6 inhibitor candidates to reduce the growth of H37Rv-GFP in human macrophages. (**a**) Intracellular H37Rv growth was assessed with high-content imaging using 1 and 10 µM doses of the ten compounds, SB415286 and C01–C09. The dotted lines show 50% and 100% (MOI1 infection control) intracellular growth of Mtb. The control groups included fixed doses of RIF + INH (1 µg/mL) or PBA (2 mM) that are highlighted as bold bars. Data in the scatter dot plot (n = 4 donors) show the median. The difference between 1 and 10 µM concentrations was determined using a two-way ANOVA and Sidak’s multiple comparisons test, while the difference between the MOI1 infection control and the test compounds was determined using a two-way ANOVA and Dunnett’s multiple comparisons test, **** *p* < 0.0001, *** *p* < 0.001, ** *p* < 0.01, * *p* < 0.05. Note that the difference between the MOI1 control and C01 (1 µM) and C03 (10 µM) was not significant (ns). (**b**) Representative microscopy images illustrating macrophages (hMDMs; light gray color) infected with H37Rv-GFP (green color) and treated with 1 or 10 µM of compound C07. Samples from whole-well images are shown at magnification ×4. Note the elongated morphology of the hMDMs at 1 µM compared to the more rounded shape of the cells at 10 µM. (**c**) Efficacy presented as the intracellular Mtb growth inhibition (%) of the controls and the test compounds, SB415286 and C01–C09, determined at 1 µM. Controls are highlighted as bold bars. Data in the bar graph (n = 4 donors) show the median +/− interquartile range and the difference in efficacy compared to untreated H37Rv-infected cells was determined using Friedman’s test, *** *p* < 0.001, * *p* < 0.05. (**d**) The efficacy of the selected compounds, SB415286, C02, and C06, to restrict the growth of H37Rv-GFP inside macrophages at 1 and 10 µM doses was assessed using CFU counts on day 3. Data in the bar graph (n = 4 donors) show the median and range. Control groups are highlighted as bold bars. *** *p* < 0.0001. (**e**) Bar graph showing the percentage uptake of H37Ra-GFP bacteria into hMDMs after 4 h of infection in the presence of the selected compounds, SB415286, C02, and C06, compared to the MOI1 infection control and RIF + INH (1 µg/mL), as evaluated by flow cytometry. One representative experiment is shown, including the median and range from n = 2 donors. (**f**) Representative microscopy images show hMDMs (light gray) infected with H37Rv-GFP (green) and treated with either control compounds (RIF + INH or PBA) or test compounds (SB415286 and C01–C09). Note that the only uninfected control condition is depicted in the first image in the upper left corner. An image of tubacin-treated cells, which induce cytotoxicity at a 10 µM dose (dying cells in red), is also included. Arrows denote H37Rv-GFP-infected cells. The magnification is ×20.

**Figure 4 biomolecules-15-00550-f004:**
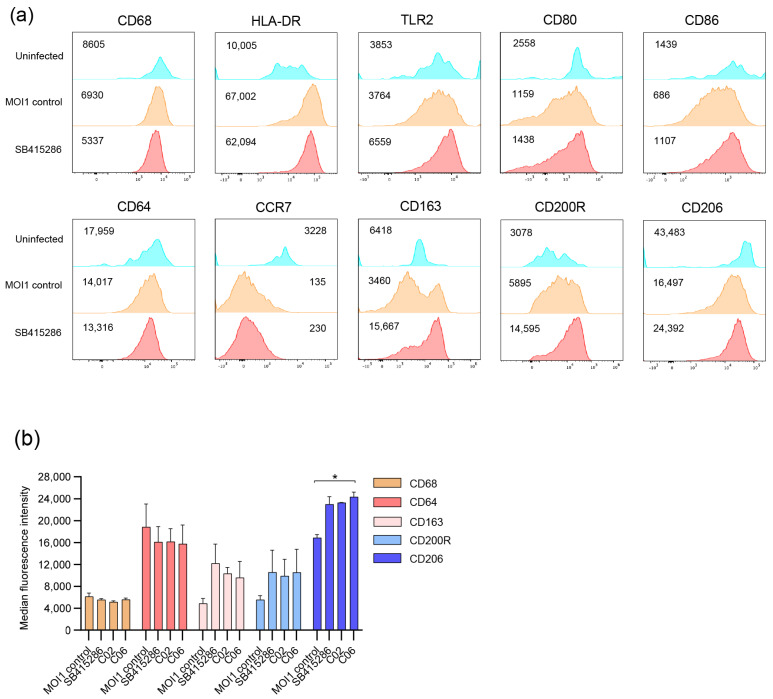
Multiparametric flow cytometry of Mtb-infected macrophages treated with SB415286 or the selected compounds C02 and C06. hMDMs were infected with H37Ra-GFP for 4 h at an MOI of 1 followed by flow cytometry staining. (**a**) Histograms showing the median fluorescence intensity of different macrophage markers on uninfected (blue), H37Ra-GFP-infected hMDM (brown), and H37Ra-GFP-infected hMDMs treated with SB415286 (red). (**b**) Bar graph showing the median fluorescence intensity of CD68, CD64, CD163, CD200R, and CD206 on H37Ra-GFP-infected hMDMs in the presence or absence of treatment with SB415286, C02, or C06. One representative experiment is shown, including the median and range from n = 2 donors. The difference between the MOI1 infection control and the test compounds was determined using a two-way ANOVA and Dunnett’s multiple comparisons test, * *p* < 0.05.

**Figure 5 biomolecules-15-00550-f005:**
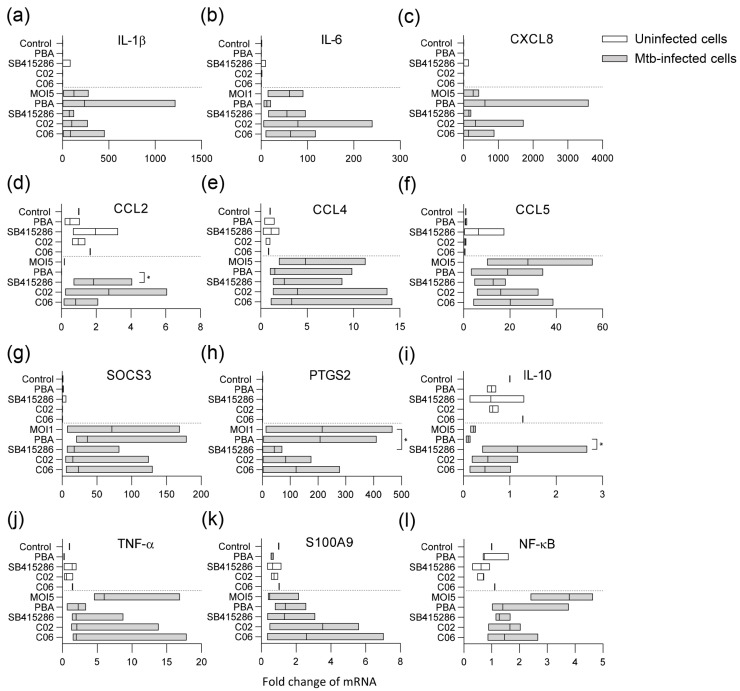
Multiplex RNA expression profiling of Mtb-infected macrophages treated with PBA, SB415286, or the selected compounds C02 and C06. Uninfected (white bars) and H37Rv-infected hMDMs (MOI5) (gray bars) were treated with the medium only (control or MOI5), PBA (2 mM), SB415286, C02, or C06 (1 µM), and assessed for mRNA expressions of (**a**) IL-1β, (**b**) IL-6, (**c**) IL-8 (CXCL8), (**d**) CCL2, (**e**) CCL4, (**f**) CCL5, (**g**) SOCS-3, (**h**) PTGS-2 (COX2), (**i**) IL-10, (**j**) TNFα, (**k**) S100A9, and (**l**) NFκB. Data assessed with Taqman array cards at 24 h from n = 3 donors are presented in floating bar graphs (median and min/max) and were analyzed using Friedman’s and Dunn’s multiple comparisons test (Mtb-infected groups only). The mRNA-fold change in uninfected and untreated cells was set to 1. * *p* < 0.05.

## Data Availability

The RNA array datasets (Ct values and fold induction) of the 23 tested genes from donors that were generated and analyzed during the current study are available online in Appendix A. All other data that support the findings of this study are available from the corresponding authors upon reasonable request.

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
