# Peer review of "Dual GSK-3β/HDAC Inhibitors Enhance the Efficacy of Macrophages to Control Mycobacterium tuberculosis Infection"

_biomolecules, 2025, doi:10.3390/biom15040550_

Round 1

Reviewer 1 Report

Comments and Suggestions for Authors

This manuscript is quite innovative for readers, as it proposes a compound with dual activity in the control of tuberculosis. However, the results appear to be very preliminary, and some results seem to be segmented. Some methodological details are required. I suggest considering a restructuring of the manuscript.

Comments for the author

Introduction

I believe it is important to include the number of TB cases reported during 2023, as well as the number of deaths.

Please review line 41: "infection requires daily treatment with multiple antibiotics for 6-9 months or longer." It is important to add the reference as this may vary by country and the established treatment regimen.

Consider developing a brief introduction to histone deacetylase (HDAC) inhibitors, as only background information on Butyrate is provided. I think there is a significant amount of literature available that justifies the use of this family of inhibitors.

Line 69: The point made here may be misinterpreted, so I think the concept of dual inhibition of GSK-3b should be elaborated upon a bit more, as it is not entirely clear.

The objective of the study should be stated more clearly, rather than just describing the results.

Materials and Methods:

Which committee approved this study?

It is necessary to clarify some points regarding the isolation of monocytes, such as differentiation time, purity percentage, and differentiation biomarkers. Also, it is unclear whether the adherent monocytes were differentiated in the T75 flasks, and what cell density was used for each assay. I believe these details should be clarified.

Why does the MOI change between different bacteria? These variations could be affecting the observed results.

Line 117: If the drugs and bacteria are used simultaneously, how can it be ensured that they do not affect bacterial internalization? A percentage of internalization should be done.

Line 122: Please review the wording.

Results:

The mention of a previous manuscript that is in preparation suggests the segmentation of results, which I believe should be included in the same study, as they are complementary and not exclusive.

The graphical representation in figure 2 shows a wide standard deviation, so I believe the sample size should be increased. Additionally, a scatter plot should be shown to observe the behavior of each patient. Regarding the statistical value P<0.002, it is not clear under which condition this difference occurs, so this should be clarified.

Throughout the text, it is mentioned that the tested drugs are HDAC inhibitors, but this has not been demonstrated, and no details are given to support this claim. Therefore, I believe the word "candidates" should be included, or evidence should be provided showing that they actually inhibit the isoform. Additionally, it is unclear whether inhibition occurs in both isoforms for all compounds, or if there are differences in this.

In figure 3, it should be indicated that some conditions are with infection and others without, as it is not clear from the axes of the figure. 

In figure 3a, although the author describes differences, it is necessary to include the relevant descriptive statistics.

Figure 3c: Indicate which concentration was used to determine efficacy for each.

Figure 4: Homogenize the number of donors. Since the proposed drugs have dual inhibition, it would be advisable to include a dual inhibition control, as both show very different behaviors when evaluating cytokines. Also, please add the corresponding axis labels.

Author Response

Ref: Submission ID, biomolecules-3454730  

Manuscript title: Dual GSK-3β/HDAC inhibitors enhance the efficacy of macrophages to control Mycobacterium tuberculosis infection

Summary: All changes in the text of the manuscript (MS) are underlined and/or strikethrough in a compare-copy of the main manuscript file (Main_Manuscript_SB_Marked_revision). Please, note that in the main manuscript, we have now included a new Figure 1b, c and d, a new Figure 3e, and a new Figure 4a-b, in response to the reviewer's questions and comments. As new experiments have been performed with flow cytometry, we have also added a new collaborator as co-author, Dr. Magda Lourda, to the manuscript. Please, see the point-by-point reply and Marked revision files for details (the Marked revision is the file referred to in the reply below).

Reviewer 1:

Summary: This manuscript is quite innovative for readers, as it proposes a compound with dual activity in the control of tuberculosis. However, the results appear to be very preliminary, and some results seem to be segmented. Some methodological details are required. I suggest considering a restructuring of the manuscript.

  1. Introduction: I believe it is important to include the number of TB cases reported during 2023, as well as the number of deaths.

Reply: We agree with the reviewer and have now included updated numbers from the WHO on TB cases, TB-related deaths, and drug-resistant TB cases for the year 2024 (WHO, Global Tuberculosis Report, 2024, reference 1 in the manuscript). Please, see the underlined text in the Introduction on page 1, line 45-46 and 48-51.

  1. Introduction: Please review line 41: "infection requires daily treatment with multiple antibiotics for 6-9 months or longer." It is important to add the reference as this may vary by country and the established treatment regimen.

Reply: We have now added a practical guideline from the Official American Thoracic Society/Centers for Disease Control and Prevention, 2016, as reference 2 in the revised manuscript. Please, see the underlined text in the Introduction on pages 1-2, line 46-48.

  1. Introduction: Consider developing a brief introduction to histone deacetylase (HDAC) inhibitors, as only background information on Butyrate is provided. I think there is a significant amount of literature available that justifies the use of this family of inhibitors.

Reply: Thank you for this suggestion, we have now extended the introduction to include the basics of HDAC proteins and HDAC inhibitors in the revised manuscript. Please, see the underlined text in the Introduction on page 2, line 60-68, also including new references 9-14.

  1. Introduction: Line 69: The point made here may be misinterpreted, so I think the concept of dual inhibition of GSK-3b should be elaborated upon a bit more, as it is not entirely clear.

Reply: This is an appreciated reflection by the referee. To avoid confusion, we have omitted the sentence starting at line 69 and instead added a new sentence that connect the introduction to HDAC inhibitors (including PBA) as well as the role of GSK-3β in inflammation: “Overall, these studies suggest that HDAC inhibitors as well as GSK-3β inhibitors could be utilized to modulate antimicrobial and inflammatory pathways involved in human diseases.” Please, see the underlined text in the Introduction of the revised manuscript on page 2, line 82-84.

  1. Introduction: The objective of the study should be stated more clearly, rather than just describing the results.

Reply: Thank you, the objective of this study was to evaluate the novel panel of dual GSK-3β and HDAC1/HDAC6 inhibitor candidates for their potential to alter growth of intracellular mycobacteria by modulating macrophage function. This has now been clarified in the last paragraph of the Introduction of the revised manuscript on page 2, line 87-97.

  1. Materials and Methods: Which committee approved of this study?

Reply: As declared in the Institutional Review Board Statement on page 16, this research was approved by the Ethical Review Board in Stockholm (EPN), Sweden (diary number: 2010/603-31/4). To clarify, we have also added the name of the review board in the Materials and Methods section. Please, see the underlined text in the revised manuscript on page 3, line 103-104.

  1. Materials and methods: It is necessary to clarify some points regarding the isolation of monocytes, such as differentiation time, purity percentage, and differentiation biomarkers. Also, it is unclear whether the adherent monocytes were differentiated in the T75 flasks, and what cell density was used for each assay. I believe these details should be clarified.

Reply: We agree with the reviewer, and we have now included a new Figure 1b and Figure 4 to clarify some of these points. In the Materials and Methods section and in Figure 1, it is described that monocytes were differentiated for 7 days using M-CSF. This is a well-established protocol used in our laboratory, and we have previously published data related to differentiation biomarkers (PMID: 32038652 and 33016941). In the revised version of the manuscript, we have now added further details on the differentiation procedure including density of PBMCs seeded in T75 flasks, viability and purity of detached monocyte-derived macrophages (new Figure 1b), and we have also added data on activation and differentiation markers on hMDMs (new Figure 4) in the revised manuscript. Please, see the underlined text in the revised manuscript on page 3 of the Materials and Methods, line 108-119, and on page 6 of the Results, line 273-285. Please, note that we also added an additional reference, PMID: 33016941 (ref 37 in the revised manuscript), to describe our previous work using this macrophage infection model.

  1. Materials and methods: Why does the MOI change between different bacteria? These variations could be affecting the observed results.

Reply: Yes, that is true. Several different MOIs have been tested during optimization and assay development in the 96- and 384-well format, respectively. Initially, we decided to use a higher MOI of 5 for the H37Ra infection in the 96-well plates, where macrophages were first infected with the bacteria and then any remaining extracellular bacteria were washed off before addition of compounds. Instead, we chose to use a lower MOI of 1 for H37Rv infection in the 384-well plates because of the small volumes in the wells and cells were not washed after Mtb infection. Accordingly, higher MOIs usually showed elevated cell death and more rapid Mtb growth rates in cells that were cultured in the 384-well plates. While these two assay set ups are slightly different, we consider it a strength to assess mycobacterial growth inhibition in the presence of the compounds using both conditions. It appears more challenging to restrict intracellular Mtb growth in macrophages infected with a higher MOI of 5 compared to an MOI of 1, which may represent a more physiologically relevant condition (PMID: 39052544). It is likely that a low MOI more closely mimics the natural infection and in vivo situation, where macrophages encounter lower numbers of bacteria. In contrast, higher MOIs may lead to an overwhelming infection, making it more difficult for macrophages to control Mtb. Nonetheless, both assay formats demonstrated high efficacy of the compounds, particularly C02 and C06, reinforcing the conclusion that dual GSK3β-HDAC1/6 inhibitor candidates effectively restrict intracellular Mtb growth in human cells. For clarification, we have now added this information in a new paragraph in the Discussion of the revised manuscript on page 14, line 509-525. In addition, we have added a new Figure 1d at page 7 to visualize the difference in infectivity of macrophages between MOI1 and MOI5.

  1. Materials and methods: Line 117: If the drugs and bacteria are used simultaneously, how can it be ensured that they do not affect bacterial internalization? A percentage of internalization should be done. Line 122: Please review the wording.

Reply: Thank you for this observation. We have now conducted a new experiment to assess whether compound C02, C06 or SB415286, could affect mycobacterial uptake. Flow cytometry performed to assess infectivity and uptake of Mtb-GFP-labeled bacteria after 4 hours infection of macrophages, demonstrated that there was no difference in bacterial uptake comparing the MOI1 infection control with compound treatment or treatment with the antibiotics control, RIF+INH. This data has been added into a new Figure 3e, on page 9-10, lines 386-389, in the revised manuscript.

For clarification, the wording in line 122 has now been changed, please see page 4, line 168-170, of the revised manuscript.

  1. Results: The mention of a previous manuscript that is in preparation suggests the segmentation of results, which I believe should be included in the same study, as they are complementary and not exclusive.

Reply: We appreciate the reviewer's input and understand that this may be the impression from an objective standpoint. However, it is important to clarify that the mentioned manuscript in preparation and the current manuscript submitted to Biomolecules represent work by two entirely independent research groups who do not have an ongoing collaboration at the moment. Specifically, the compound structures and IC50 values were generated by the following principal investigators (PIs): Prof. Ke Ding from the Shanghai Institute of Organic Chemistry, Chinese Academy of Sciences; Prof. Mingtao Li from Sun Yat-sen University; and Prof. Weidong Le from The First Affiliated Hospital of Dalian Medical University.

Currently, these PIs are preparing a manuscript to present the biochemical structure and IC50 values for this panel of dual GSK-3β/HDAC inhibitors and their effect on Alzheimer's disease markers. We are not involved in this manuscript. Conversely, our focus is on assessing the potential of these compounds to ameliorate Mtb infection, which does not involve the listed PIs. While the IC50 values of the enzyme inhibitors are relevant for compound efficacy, the overall aims and scope of these two manuscripts are different, involving non-overlapping authors and contributors. To avoid confusion, we have updated the text from “manuscript in preparation” to “unpublished observations”, on page 6, line 263.

  1. Results: The graphical representation in figure 2 shows a wide standard deviation, so I believe the sample size should be increased. Additionally, a scatter plot should be shown to observe the behavior of each patient. Regarding the statistical value P<0.002, it is not clear under which condition this difference occurs, so this should be clarified.

Reply: We agree with the referee that it may be clearer to show the data including individual data plots, so we have now changed the graphs in Figure 2a-b, which now show scatter plots with bars (median) instead of floating bars (median and min-max). To provide a clean overview, we prefer to show the summary of efficacy in Figure 2c as grouped data.

It is our experience that using primary cells from human donors normally results in large donor-to-donor variations, including differences in the relative growth rates of Mtb within in vitro differentiated macrophages, and occasionally large inter-experimental variations. As a result, the responsiveness of Mtb-infected host cells to epigenetic modulators like GSK-3β-HDAC inhibitors may also vary. Nonetheless, we believe that using primary cells from human donors offers a significant advantage over cell lines, as it reflects the natural variation within a human population. At this stage, we do not anticipate that including additional donors will significantly change these results.

In Figure 2a, we used a Wilcoxon matched-pairs signed rank test to compare the mean values between 1 and 10 µM concentrations of each condition or compound treatment. Thus, the p-value of 0.002 indicates the overall difference overall between in 1 and 10 µM at the group level. However, as the data set passed a normality test, we have also performed a two-way ANOVA to test the individual differences between compound treatments at different doses. Using Sidak's multiple comparisons test, we found a significant difference between 1 and 10 µM comparing compound C02 (P = 0.026) and C06 (P = 0.0057), which has now been added to the graph. Additionally, we observed a significant difference between the MOI1 infection control and 10 µM of C02 (P =0.0041) and 10 µM of C06 (P =0.0004).  These significant differences are now visualized in Figure 2a and described in the Results section of the revised manuscript on page 8, line 306-309.

  1. Results: Throughout the text, it is mentioned that the tested drugs are HDAC inhibitors, but this has not been demonstrated, and no details are given to support this claim. Therefore, I believe the word "candidates" should be included, or evidence should be provided showing that they actually inhibit the isoform. Additionally, it is unclear whether inhibition occurs in both isoforms for all compounds, or if there are differences in this.

Reply: We thank the reviewer for this insightful comment. This panel of dual inhibitor compounds was generated based on their individual capacity to inhibit either GSK-3β, HDAC1 and/or HDAC6 proteins in cell-free assays. Please, see the table in the supplementary file, “IC50 values of compounds” that has been attached for review purpose only. In this table, the respective IC50 value for each of the three different enzymes has been listed (unpublished data belonging to another research group). This has been clarified in the revised manuscript text by including the range of the IC50 values of each enzyme and compound as well as their specificity. Please, see the revised text on page 6, line 263-267. According to the suggestion from the reviewer, we use the phrase “dual GSK-3β/HDAC inhibitors candidates” or “test compounds” throughout the manuscript text.

  1. Results: In figure 3, it should be indicated that some conditions are with infection and others without, as it is not clear from the axes of the figure.

Reply: Please, note that all conditions in Figure 3, except the first image in Figure 3f, include Mtb-infected cells. The control image in Figure 3f, has been clearly labeled as “Uninfected”, and we have now added to the Figure legend that the only uninfected control condition is depicted in the first image at the upper left in Figure 3f.

  1. Results: In figure 3a, although the author describes differences, it is necessary to include the relevant descriptive statistics.

Reply: Similar to Figure 2a-b, we now show the data in Figure 3a in a scatter plot with bars (median) instead of floating bars (median and min-max). Statistical differences were calculated using a two-way ANOVA and Dunnett's s multiple comparisons test as visualized in Figure 3a on page 10, and line 368-370 on page 9 of the revised manuscript.

  1. Results: Figure 3c: Indicate which concentration was used to determine efficacy for each.

Reply: Thank you for this comment, and as indicated by the bar color in the figures, Figure 2c show H37Ra-data from 10 µM concentrations (light grey bars), while Figure 3c show H37Rv-data from 1 µM concentration (white bars). To clarify this, we have now added a legend with the bar color and concentration to the graphs in Figure 2c and 3c in the revised manuscript on pages 8 and 10.

  1. Results: Figure 4: Homogenize the number of donors. Since the proposed drugs have dual inhibition, it would be advisable to include a dual inhibition control, as both show very different behaviors when evaluating cytokines. Also, please add the corresponding axis labels.

Reply: We have now homogenized the number of donors and also added legends to the bars and X-axis. This RNA array was performed as part of one large experiment with several donors. Data was measured as fold change of mRNA relative to the uninfected and untreated control. This has now been added to the X-axis. We have also added a legend to the bar graphs showing uninfected vs Mtb-infected cells and these sub-groups are now separated by a thin dotted line in each graph, please see page 13 of the revised manuscript.

Reviewer 2 Report

Comments and Suggestions for Authors

In this manuscript, the authors tested the anti-Mycobacterium tuberculosis (Mtb) growth inhibition efficiencies of small chemical inhibitors targeting both glycogen synthase kinase 3 beta (GSK-3β) and histone deacetylases (HDAC) in human monocyte derived macrophages infected with two lab adapted strains of Mtb, H37Ra expressing GFP and H37Rv, using a high content live cell imaging by IncuCyte and CFU counting  to monitor the growth of Mtb strains in macrophages pretreated with  nine different compounds which target both GSK-3β and HDAC1 and 6 simultaneously.  The control chemicals included GSK-3β specific inhibitor SB415286 and HDAC inhibitor and pan HDAC inhibitor phenylbutyrate (PBA) and combination of two of the standard anti-Mtb drugs, isoniazid and rifampin, as positive control. The preliminary data showed that two out of nine tested compounds, namely compounds 2 and 6, showed strong inhibitory effects on the growth of the Mtb strains in human MDMs without affecting the viability of the cells though all the other compounds showed some level of Mtb growth inhibition effects in MDMs. The regulation of inflammatory cytokine, chemokines and other effector molecules in Mtb infected MDMs were also assessed at the mRNA levels as determined by custom designed multiplex mRNA expression array. Overall, this study showed the anti-Mtb effects of the novel compounds which target both GSK-3βand HDAC1 and 6, in a human primary MDMs though without any significant superiority in their efficiencies as compared to the chemicals that target more specific molecules in macrophages, such as GSK-3β or HDACs. The manuscript was written well, the experimental methods were detailed enough, and the data were presented reasonably well and support the conclusions. However, there are some minor concerns which require revision of the manuscript before it is considered for publication.

Concerns:

1.      Although it was stated that the human monocyte derived macrophages (MDMs) from CD14+ cells were used for this study, there is no evidence for the confirmation of these cells as CD14+ cells and MDMs before infected with Mtb.

2.      Figure 1a, though it was labeled as p < 0.002, it is not clear which compound showed significantly better Mtb inhibitory effects and compared with what? However, Fig. 1c showed only two compounds, compound 2 and 6, had significantly better H37Ra inhibitory effects, which is confusing.  

3.      It is not clear why three days used for imaging studies and which studies were performed after 5-day infection as experimental study outline stated as 3 or 5 days and the 5 days showed any better efficiencies for the chemicals. This should be clearly stated in the methods, results or in the figure legend.

4.      Figure 4, thought the figure legend stated as MOI5, but the figures show MOI1, which is the correct infection dose? For all the graphs in this figure, there is no label for the numbers in the X axes, such as expression levels in arbitrary units or fold change over non infected control cells. It would also be clearer and easier to follow if the upper five bars were labelled as control cells without Mtb infection and the lower five bars as Mtb infected cells. In figure legend, it was stated that n=2-5, and this can also be specified for each panel of the figure for the clarity.

5.      Although discussion is a bit long, there is no discussion of potential causes for the compounds with dual targets, targeting both GSK and HDAC, did not show better efficiencies than that of the GSK inhibitor or HDAC inhibitor and how this could be pursued in the future studies.  

Author Response

Ref: Submission ID, biomolecules-3454730  

Manuscript title: Dual GSK-3β/HDAC inhibitors enhance the efficacy of macrophages to control Mycobacterium tuberculosis infection

Summary: All changes in the text of the manuscript (MS) are underlined and/or strikethrough in a compare-copy of the main manuscript file (Main_Manuscript_SB_Marked_revision). Please, note that in the main manuscript, we have now included a new Figure 1b, c and d, a new Figure 3e, and a new Figure 4a-b, in response to the reviewer's questions and comments. As new experiments have been performed with flow cytometry, we have also added a new collaborator as co-author, Dr. Magda Lourda, to the manuscript. Please, see the point-by-point reply and Marked revision files for details (the Marked revision is the file referred to in the reply below).

Reviewer 2:

Summary: In this manuscript, the authors tested the anti-Mycobacterium tuberculosis (Mtb) growth inhibition efficiencies of small chemical inhibitors targeting both glycogen synthase kinase 3 beta (GSK-3β) and histone deacetylases (HDAC) in human monocyte-derived macrophages infected with two lab adapted strains of Mtb, H37Ra expressing GFP and H37Rv, using a high content live cell imaging by IncuCyte and CFU counting to monitor the growth of Mtb strains in macrophages pretreated with nine different compounds which target both GSK-3β and HDAC1 and6 simultaneously. The control chemicals included GSK-3βspecific inhibitor SB415286 and HDAC inhibitor and pan HDAC inhibitor phenylbutyrate (PBA) and combination of two of the standard anti-Mtb drugs, isoniazid and rifampin, as positive control. The preliminary data showed that two out of nine tested compounds, namely compounds 2 and 6, showed strong inhibitory effects on the growth of the Mtb strains in human MDMs without affecting the viability of the cells though all the other compounds showed some level of Mtb growth inhibition effects in MDMs. The regulation of inflammatory cytokine, chemokines and other effector molecules in Mtb-infected MDMs were also assessed at the mRNA levels as determined by custom designed multiplex mRNA expression array. Overall, this study showed the anti-Mtb effects of the novel compounds which target both GSK-3βand HDAC1 and 6, in human primary MDMs though without any significant superiority in their efficiencies as compared to the chemicals that target more specific molecules in macrophages, such as GSK-3β or HDACs. The manuscript was written well, the experimental methods were detailed enough, and the data were presented reasonably well and support the conclusions. However, there are some minor concerns which require revision of the manuscript before it is considered for publication.

  1. Although it was stated that the human monocyte derived macrophages (MDMs) from CD14+ cells were used for this study, there is no evidence for the confirmation of these cells as CD14+ cells and MDMs before infected with Mtb.

Reply: We thank the reviewer for this comment, and we have now included new experiments using flow cytometry to show the percentage of CD14-positive cells among bulk peripheral blood mononuclear cells (PBMCs). We have also stained differentiated monocyte-derived macrophages for the macrophage marker CD68 as well as CD14. These results demonstrated >99% viable cells and that approximately 60% of MCSF-treated cells became fully differentiated, while remaining cells were single-positive for CD14 eg. not fully differentiated monocytes, or CD68-negCD14-neg cells that were likely lymphocytes. These results are now presented in a new Figure 1b-d in the revised manuscript at page 7 and described in the Result section on page 6, line 273-285.

  1. Figure 1a, though it was labeled as p < 0.002, it is not clear which compound showed significantly better Mtb inhibitory effects and compared with what? However, Fig. 1c showed only two compounds, compound 2 and 6, had significantly better H37Ra inhibitory effects, which is confusing.

Reply: This reflection is correct. In Figure 2a, we used a Wilcoxon matched-pairs signed rank test to compare the mean values between 1 and 10 µM concentrations of each condition or compound treatment. Thus, the p-value of 0.002 indicates the overall difference overall between in 1 and 10 µM at the group level. However, as the data of the test panel passed a normality test, we have also performed a two-way ANOVA to test the individual differences between compound treatments at different doses. Using Sidak's multiple comparisons test, we found a significant difference between 1 and 10 µM comparing compound C02 (P = 0.026) and C06 (P = 0.0057), which has now been added to the graph. Additionally, we observed a significant difference between the MOI1 infection control and 10 µM of C02 (P =0.0004) and 10 µM of C06 (P =0.0041).  These significant differences are now visualized in Figure 2a and described in the Results section of the revised manuscript on page 8, line 306-309. The results are in line with the efficacy data shown in Figure 2c.

  1. It is not clear why three days used for imaging studies and which studies were performed after 5-day infection as experimental study outline stated as 3 or 5 days and the 5 days showed any better efficiencies for the chemicals. This should be clearly stated in the methods, results or in the figure legend.

Reply: We understand that the timing and rationale for certain time-points used in high-content imaging analyses may not be immediately clear. As indicated in the revised experimental outline in Figure 1 and described in the Materials and Methods section 2.6, the readout of live-cell imaging varied between 3 to 5 days. This variation depended on the rate of intracellular mycobacterial growth in the MOI-infection control and the integrity of Mtb-infected macrophages in culture, which was monitored every 12 hours using live-cell imaging. If the bacteria were growing rapidly in cells from a given donor, the experiment was concluded on day 3. Conversely, if the bacteria were growing slowly in the cells, the experiment could be extended until day 5 before the final readout. We have now tried to clarify this in section 2.6 of the Materials and method section of the revised manuscript, page 4-5, line 197-201.

  1. Figure 4, thought the figure legend stated as MOI5, but the figures show MOI1, which is the correct infection dose? For all the graphs in this figure, there is no label for the numbers in the X-axes, such as expression levels in arbitrary units or foldchange over non infected control cells. It would also be clearer and easier to follow if the upper five bars were labelled as control cells without Mtb infection and the lower five bars as Mtb-infected cells. In figure legend, it was stated that n=2-5, and this can also be specified for each panel of the figure for the clarity.

Reply: This is an appreciated reflection by the referee. MOI5 was correct for this experiment and data was measured as fold change of mRNA relative to the uninfected and untreated control. This has now been added to the X-axis. We have also added a legend to the bar graphs showing uninfected vs Mtb-infected cells and these sub-groups are now separated by a thin dotted line. Finally, we have homogenized the number of donors, please see Figure 5 and the legend on page 13 in the revised version of the manuscript.

  1. Although discussion is a bit long, there is no discussion of potential causes for the compounds with dual targets, targeting both GSK and HDAC, did not show better efficiency than that of the GSK inhibitor or HDAC inhibitor and how this could be pursued in the future studies.

Reply: The referee correctly noted that the difference between the various dual inhibitors and the GSK-3β inhibitor scaffold, SB415286, was not significant. However, the efficacy assessed using H37Ra-infected cells (Figure 2c) and H37Rv-infected cells (Figure 3c) demonstrated that only compounds C02 and C06 exhibited significant potency in reducing intracellular Mtb growth at doses of 10 µM and 1 µM, respectively. It is also important to remember that these molecules were initially developed to treat neurological disorders. As described in the Discussion (line 587-607), we recently discovered that HDAC inhibitors specifically targeting Sirtuin 2 show promise in modulating the macrophage response to reduce intracellular Mtb growth. In this context, we anticipate that a combination of dual GSK-3β and Sirtuin 2 inhibitors will be highly relevant for future investigations as host-directed therapy for TB.

Round 2

Reviewer 1 Report

Comments and Suggestions for Authors

Thank you very much for taking the time to address each of the observations. From my perspective, the study is now much more comprehensive and could be considered for publication